# The Intricate Interplay between Cell Cycle Regulators and Autophagy in Cancer

**DOI:** 10.3390/cancers14010153

**Published:** 2021-12-29

**Authors:** Dorian V. Ziegler, Katharina Huber, Lluis Fajas

**Affiliations:** Center for Integrative Genomics, University of Lausanne, CH-1015 Lausanne, Switzerland; katharina.huber@unil.ch

**Keywords:** cell cycle regulators, autophagy, CKI, CDKs, E2F, cancer

## Abstract

**Simple Summary:**

Autophagy is an intracellular catabolic program regulated by multiple external and internal cues. A large amount of evidence unraveled that cell-cycle regulators are crucial in its control. This review highlights the interplay between cell-cycle regulators, including cyclin-dependent kinase inhibitors, cyclin-dependent kinases, and E2F factors, in the control of autophagy all along the cell cycle. Beyond the intimate link between cell cycle and autophagy, this review opens therapeutic perspectives in modulating together these two aspects to block cancer progression.

**Abstract:**

In the past decade, cell cycle regulators have extended their canonical role in cell cycle progression to the regulation of various cellular processes, including cellular metabolism. The regulation of metabolism is intimately connected with the function of autophagy, a catabolic process that promotes the efficient recycling of endogenous components from both extrinsic stress, e.g., nutrient deprivation, and intrinsic sub-lethal damage. Mediating cellular homeostasis and cytoprotection, autophagy is found to be dysregulated in numerous pathophysiological contexts, such as cancer. As an adaptative advantage, the upregulation of autophagy allows tumor cells to integrate stress signals, escaping multiple cell death mechanisms. Nevertheless, the precise role of autophagy during tumor development and progression remains highly context-dependent. Recently, multiple articles has suggested the importance of various cell cycle regulators in the modulation of autophagic processes. Here, we review the current clues indicating that cell-cycle regulators, including cyclin-dependent kinase inhibitors (CKIs), cyclin-dependent kinases (CDKs), and E2F transcription factors, are intrinsically linked to the regulation of autophagy. As an increasing number of studies highlight the importance of autophagy in cancer progression, we finally evoke new perspectives in therapeutic avenues that may include both cell cycle inhibitors and autophagy modulators to synergize antitumor efficacy.

## 1. Introduction

The cell cycle constitutes a fundamental biological process divided into four successive steps: G0/G1, S, G2, and M phases. Cell-cycle progression is tightly regulated by both extrinsic (e.g., nutrient quantity, presence of growth factors) and intrinsic (e.g., cell size, macromolecular damage) cues [1]. Downstream, cell-cycle regulators integrate these signals to adapt a pro- or anti-proliferative response. As crucial effectors of cell-cycle progression, cell-cycle regulators participate in a myriad of pathophysiological processes, including development, tissue regeneration, and cancer [2]. These cell-cycle effectors include the cyclin-dependent kinase (CDKs) holoenzymes, which contain and are activated by the regulatory subunits, the cyclins. When active, these CDKs phosphorylate multiple substrates, including the retinoblastoma protein (pRB), ultimately resulting in the release of the E2F transcription factors and the transcription of genes involved in cell cycle progression and cell division [3] (Figure 1a). The activity of CDKs is inhibited by cyclin-dependent kinase inhibitors (CKIs), thus arresting the cell cycle. The cyclins/CDK-pRB-E2F pathway is commonly deregulated during tumorigenesis, making CDK inhibitors valuable and promising antitumoral therapeutic tools [4,5].

Beyond their canonical role, cell cycle regulators also control cellular metabolic functions, especially in the cancer context [6]. Conversely, important metabolic and glycolytic enzymes, including pyruvate dehydrogenase kinase (PDK) [7], pyruvate dehydrogenase [8], glyceraldehyde-3-phosphate dehydrogenase (GAPDH) [9], and pyruvate kinase M2 (PKM2) [10] possess non-canonical roles in the regulation of cell-cycle progression [11]. As cancer cells need to adapt to an increased demand for growth and proliferation, metabolic reprogramming is a key hallmark of tumor cells, offering several advantages to highly proliferating cells, concerning both bioenergetics and biosynthetic requirements [12]. In this setting, the importance of cell-cycle regulators in coupling cell proliferation and intracellular metabolism has been recently unraveled [6,13,14].

Upon the receipt of stress signals, autophagy is a catabolic process allowing the removal and recycling of metabolic intermediates. Autophagy is divided into two independent complementary mechanisms. First, selective autophagy through micro- or chaperone-mediated autophagy (CMA), and second, non-selective bulk autophagy, termed macroautophagy [15]. Many advanced stage cancers have increased autophagic activity, which could reflect a pro-survival adaptation to extrinsic stress [16]. Autophagy was found to modify the fate of cancer cells, promoting in some cases cellular senescence and resistance to cell death. In line with this, systemic or cancer-specific inhibition of autophagy dampens the progression of many cancers [16]. Thus, this autophagic process is necessary for the survival of many cancer subtypes, including RAS and BRAF-driven tumors [16]. Notably, host autophagy is also crucial in modulating the tumor microenvironment, with an opposite effect on cancer progression. On the one hand, the immune system autophagy has an antitumoral effect because it ensures tumor recognition and elimination [17]. On the other hand, host autophagy could also be pro-tumoral, as demonstrated by increased levels of circulating arginine, which is essential for tumor growth [18]. Taken together, these data suggest a critical role for autophagy in tumor promotion, supported by multiple clinical trials in sarcoma, melanoma, pancreatic cancer, and glioblastoma, demonstrating the potential benefit of the clinical inhibition of autophagy [16]. Nonetheless, the current view of the tumor-promoting role of autophagy was also revised with the discovery of an opposite role of autophagy as a tumor suppressor mechanism in early tumor development, which could notably facilitate the degradation of oncogenic molecules or enhance cancer immunosurveillance [17,19]. Collectively, these data underline the fact that the dual role of autophagy during tumorigenesis is highly context-dependent [15].

At the molecular level, macroautophagy requires the creation of autophagosomes (the nucleation phase) and their subsequent fusion with lysosomes (the elongation phase) to ultimately form autolysosomes. The initiation step involves the formation of the ULK1/ATG1 complex, later directly phosphorylating a key component of the Class III PI3K complex, VPS34. The further association of this Class III PI3K complex, notably with UVRAG, ATG14, and Beclin-1, allows the proper nucleation of phagophores vesicles. The incorporation of organelles or targeted proteins are mediated by chaperones, including p62 (Figure 1b). In the next step, the concomitant action of ATG4B and ATG7 generates mature LC3II, later incorporating into growing phagophore external membrane (Figure 1b). Finally, the fusion of autophagosome with lysosome leads to the terminal degradation of engulfed materials (Figure 1b).

Autophagy is controlled by multiple signaling cues that are notably integrated by the main intracellular sensor of cellular stress and growth factor stimuli, i.e., the mammalian target of rapamycin (mTOR). Via AKT or MAPK signaling, the activation of mTOR reduces macroautophagy by inhibiting the ULK1/ATG1 kinase. In contrast, the downregulation of mTOR by AMPK or p53 signaling activates ULK1/ATG1 kinase promoting autophagy. During the progression of the cell cycle, autophagy mostly occurs during interphase, in particular in the G1 and S phases [20,21,22]. Indeed, repression of autophagy during mitosis protects condensed genomes and organelles movements from accidental autophagic engulfment. Hence, there is a required control of autophagy by the cell-cycle regulators. An increasing number of studies confirmed this hypothesis and unraveled additional non-canonical roles for cell-cycle regulators. [23,24,25].

In this review, we will describe how main cell-cycle regulators, particularly in focusing on cyclin-dependent kinase inhibitors (CKIs), CDKs-cyclins complexes, and E2F transcription factors, impact key steps of the autophagic process. In addition, how autophagy impacts cell-cycle regulators, notably modulating their quantity all along the cell cycle, will be addressed. Finally, the idea of a therapeutic avenue combining both cell-cycle and autophagy modulators during tumorigenesis will be examined.

## 2. Cell-Cycle Regulators Modulate Autophagy

Cell-cycle regulation integrates both external and internal stimuli to sustain cellular viability. Multiple regulators of the cell cycle, including (among others) CKI, CDKs, and E2F transcription factors, were found to regulate autophagic processes. 

### 2.1. Cyclin-Dependent Kinase Inhibitors and Autophagy

CKIs constitute crucial regulators of cell-cycle progression, as they directly inhibit the CDKs’ activities that mediate the mitotic checkpoints. Acting to mediate cell-cycle arrest transiently or permanently depending on contexts, CKIs act as tumor suppressors and most CKI-deleted mice are prone to develop multiorgan hyperplasia [26]. Among them, p27^Kip1^, p21^CIP1^, p57^CIP2^, and p16^INK4A^ (Figure 1a) were shown to regulate some features of autophagy.

#### 2.1.1. The CIP/KIP Family and Autophagy

The protein p27^KIP1^ belongs to the CIP/KIP family of CKI inhibiting CDK1-CDK2/cyclins A-B and CDK4-6/cyclin D complexes (Figure 1a) and is mostly activated by extracellular antimitogenic signals [27]. For example, serum starvation and cell density promote its expression, ultimately resulting either in cell-cycle arrest in G1 or apoptosis. Importantly, the subcellular localization of p27^KIP1^ dictates its function and displays apparent opposite roles. On the one hand, the tumor suppressor role of p27^KIP1^ is primarily associated with its nuclear form-enhancing cell-cycle inhibition and sensitivity to cellular senescence and apoptosis [28]. On the other hand, cytoplasmic p27^KIP1^ drives survival mechanisms, notably through the activation of autophagy [29]. Indeed, constitutive expression and accumulation of p27^KIP1^ induce autophagy in glioma cells and myeloma [28,30]. Mechanistically, p27^KIP1^ activation of autophagy has been linked to its phosphorylation at Thr298, notably by the activation of the LKB1/AMPK pathway upon amino acid starvation [31,32]. Activated p27^KIP1^ is recruited to lysosomes, where it interacts with LAMTOR1 to prevent mTORC1 activation, ultimately resulting in the induction of autophagy [25] (Figure 2a). It is noteworthy that p27^KIP1^ mediates the survival of cells through autophagy. Thus, it is at the crossroads of the cell fate decision, including cell death. Indeed, while the absence of p27^KIP1^ induces caspase-dependent apoptosis, its accumulation triggers cell death through both caspase-independent and autophagy-dependent mechanisms in multiple cancer cells [28,30,31]. Altogether, balanced nuclear and cytoplasmic p27^KIP1^ levels appear necessary to mediate optimal tumor cell survival. 

Among the CIP/KIP family, p21^CIP1/WAF1^ also inhibits multiple cyclin-CDK complexes, including cyclin E-CDK and cyclin A/B-CDK1, stopping cell-cycle progression at the G1 and S phases [33] (Figure 1a). In contrast to p27^KIP1^, p21^CIP1/WAF1^ is mostly activated by stress-signaling, such as irradiation or pro-apoptotic stresses, and is a direct target of the well-known tumor suppressor p53. Therefore, p21^CIP1/WAF1^ is not only linked to the arrest of cell proliferation; it is also associated with cellular senescence and apoptosis, depending on the nature, intensity, and duration of the stress [34]. The sole p21^CIP1/WAF1^ ectopic expression functionally induces both autophagy and cellular senescence in triple-negative breast cancer cells [35] (Figure 2a). Multiple stresses, including oxidative stress, link increased p21^CIP1/WAF1^ expression with the induction of autophagy [36,37,38,39,40]. Mechanistically, p21^CIP1/WAF1^-induced autophagy seems to rely on ATG5 stabilization in fibroblasts [36] and AMPK phosphorylation [39], or LC3B interaction [40], in cardiomyocytes. Nonetheless, whether p21^CIP1/WAF1^ may directly modulate the activity of autophagy key players in tumoral contexts remains to be addressed and confirmed in future investigations. Finally, and according to these studies, the pro-autophagic role of p53 (comprehensively reviewed elsewhere) [41] through the inhibition of mTORC1 complex may rely partially on the contribution of p21^CIP1/WAF1^.

The last CKI of the CIP/KIP family is p57^CIP2^. Only one study reported a link between p57^CIP2^ and autophagy. In contrast to p27^KIP1^ and p21^CIP1/WAF1^, p57^CIP2^ accumulation decreases autophagy in hepatocarcinoma cells upon EGFR-targeted therapy, through upregulation of the PI3K/AKT/mTOR axis [42]. Nonetheless, whether the sole expression of p57^CIP2^ may impact autophagy is unknown. Taken together, these results demonstrate that the CIP/KIP family of CKI, specially p27^KIP1^ and p21^CIP1/WAF1^*,* has an overall pro-autophagic role (Figure 2a).

#### 2.1.2. The INK Family and Autophagy

The protein p16^INK4A^ is a member of the INK family and inhibits the association of cyclin 4/6 with cyclin D, arresting cell-cycle progression at the G1 phase [43] (Figure 1a). Similar to p21^CIP1^, the expression of p16^INK4A^ induces both senescence and autophagy in triple-negative breast cancer cells [35]. This pro-autophagic effect of p16^INK4A^ was shown to be RB-dependent in glioblastoma [44] and could be partially explained by the transcriptional regulation of autophagic genes by the E2F transcription factors (see the subsection “”). More recently, some studies emphasized that the autophagic process regulates p16^INK4A^ quantity and localization, indicating crucial cross-talks and feedbacks between the cell-cycle regulators and autophagy [45,46,47]. Interestingly, the role of p16^INK4A^ in autophagy seems to be interrelated with its role during cellular senescence [48]. Finally, the role of other INK family CKI members, namely p15^INK4B^, p18^INK4C^, and p19^INK4D^, remains unknown in the modulation of autophagy.

### 2.2. Cyclin-Dependent Kinases and Autophagy

Cyclin-dependent kinases (CDKs) are direct targets of CKIs and, when associated with distinct cyclins, constitute the main effector kinases regulating key processes during all phases of the cell cycle. They include two distinct subfamilies: cell-cycle-associated CDKs (CDK1, 2, 4, and 6), which directly participate in the cell-cycle progression; and transcription-associated CDKs (CDK7, 8, 9, 11, 12, and 13), which directly participate in the regulation of gene transcription. Importantly, most of the studies highlighted the role of cell-cycle-associated CDKs in the control of autophagy.

#### 2.2.1. S/G2/M Phases CDKs in the Control of Autophagy

CDK1 is a kinase associated with cyclin A in the G2 phase and with cyclin B during the mitosis phases of the cell cycle (Figure 1a). CDK1 has been linked to macroautophagy and to multiple autophagic key players. Indeed, CDK1 phosphorylates the Thr159 of VPS34, and inhibits the interaction of the latter with Beclin-1, subsequently blocking the initial steps of autophagy nucleation [49]. Furthermore, CDK1 phosphorylates p62 at Thr269 and Ser272, thus stabilizing cyclin B1 levels [50]. An additional mechanism of the anti-autophagic role of CDK1 was demonstrated by two independent studies [24,51]. The protein complex mTORC1 represses the autophagic process in the interphase of the cell cycle; however, mTORC1 is inhibited during mitosis, and therefore cannot repress autophagy at this phase. In a similar way to mTORC1, mitotic CDK1/cyclin B1 phosphorylates and thus inhibits multiple autophagic key players, including ULK1, ATG13, ATG14, and TFEB [24,51]. In addition to its role in macroautophagy, CDK1 limits the chaperone-mediated autophagy of HIF1-α, facilitating cell-cycle progression under hypoxic conditions [52]. Taken together, these data indicate that CDK1 inhibits autophagy through phosphorylation of multiple key autophagic effectors. This CDK1-mediated inhibition of autophagy appears to be necessary to avoid the accidental engulfment of condensed genomes or organelles during critical steps of mitosis.

CDK2 is another key kinase associated with cyclin A or E and is mostly involved in late G1/S interphase and in G2 (Figure 1a). CDK2 has been not directly shown to play a role in autophagy induction, but genetic manipulation of p27^KIP1^ through miR-221 demonstrates the functional involvement of CDK2 in inhibiting autophagy in cardiomyocytes. Mechanistically, the inhibition of CDK2 restores p27^KIP1^ and mTORC1 mediates autophagy. Interestingly, like CDK1, CDK2 also promotes the lysosomal degradation of HIF-1α at the G1/S transition phase, allowing cell-cycle progression under hypoxic conditions [52]. Altogether, these data also underscore the anti-autophagic role of CDK2.

#### 2.2.2. G1 Phase CDKs in the Control of Autophagy

CDK4/6 associate with cyclin D to phosphorylate pRB and release E2F transcription factors during the G1 phase (Figure 1a). Beyond the Rb-dependent role, many other substrates are phosphorylated by CDK4 or CDK6 [14]. CDK4/6 were most recently studied in the context of autophagy and cellular senescence, through the use of the multiple chemical inhibitors available, including (among others) abemaciclib, palbociclib, or ribociclib [4]. 

CDK4/6 inhibitors (CDK4/6i) also induce cellular senescence in cancer cells [53] promoting in parallel autophagy [54,55,56,57]. Mechanistically, CDK4/6 inhibits senescence through the phosphorylation and stabilization of DNMT1 [54]. CDK4/6i may thus mediate senescence through autophagy-dependent degradation of DNMT1 [54]. Similarly, palbociclib also induces autophagy and cellular senescence in Rb-positive cytoplasmic cyclin E negative cancers [56]. Two recent studies investigated the importance of CDK4/6 coupling cell-cycle progression and the cell growth of breast cancer cells [57,58]. cyclinD-CDK4/6 regulates both the activation and localization of mTORC1.

On the one hand, CyclinD-CDK4/6 negatively regulates TSC2 through phosphorylation of Ser1217 and Ser1452, subsequently activating mTOR [58]; on the other hand, it phosphorylates folliculin to enhance its recruitment to the lysosomal surface upon amino acids deprivation [57]. Finally, a recent study demonstrates the importance of CDK4/6 in the inhibitory regulation of lysosome biogenesis through TFEB/TFE3 phosphorylation [59]. Taken together, these data indicate that cyclinD-CDK4/6 is crucial in promoting mTORC1 activation and localization and determinant in negatively regulating lysosome biogenesis (Figure 2b). The use of CDK4/6i counteracts these actions and leads ultimately to impaired autophagic flux [57,58]. 

#### 2.2.3. Non-Canonical CDKs and Autophagy

Only a few studies reported a role in autophagic process for non-canonical CDKs, in particular, the roles of CDK5 and CDK11. Not typically activated by cyclins, CDK5 phosphorylates pRB, ultimately resulting in the progression of the cell cycle [60]. CDK5 was documented for its pro-autophagic function, especially in the neuronal context [61,62,63,64,65]. Macroautophagy is thus universally enhanced by CDK5 kinase activity in mice, monkeys, and flies [61,63,64]. CDK5 mediates the phosphorylation of Acinus at Ser437, promoting basal autophagy in the fly brain [64]. Endophilin B1 is also phosphorylated by CDK5 to sustain neuronal autophagy upon starvation [61]. In chaperone-mediated autophagy, CDK5 has a dual role. On the one hand, CDK5 phosphorylates RKIP at Thr42 to promote its chaperone-mediated autophagy through Hsc70 [62]. On the other hand, CDK5 promotes the phosphorylation-mediated degradation of BAG3 [65], one key component of the HSP70-BAG3-mediated autophagy machinery [66]. In addition to brain-related studies, it would be of interest for future studies to address the role of CDK5 during autophagy in the cancer context. Concerning CDK11, which is mostly activated by L-type cyclins [67], only an anti-autophagic role has been reported. CDK11 knockdown induced autophagy according to accumulated LC3II levels in breast cancer cells [68]. 

Collectively, and beyond the canonical cell-cycle-associated CDKs (CDK1, 2, 4, and 6), these studies also suggest a relative and neglected role of non-canonical CDKs in the modulation of autophagic process.

#### 2.2.4. CDKs and Mitophagy

Mitophagy is a selective autophagy process, involving sequestration of damaged dysfunctional mitochondria to maintain global mitochondrial homeostasis [69]. Mitochondria fragments are more sensitive to mitophagy, and mitochondrial fission is necessary to initiate proper mitophagy processing [69]. The importance of cell-cycle regulators in mitochondrial dynamics emerged with the various observations of hyperfused mitochondria in genetically and pharmacologically alterations of CDK and cyclins. For instance, cyclin D1 −/− mammary gland adipocytes [70] or CDK4/6 inhibition in pancreatic cancer cells [13] display hyperfused mitochondria. Interestingly, the cyclinB-CDK1 complex promotes the mitotic phosphorylation of DRP1, a key fission protein [71]. Nevertheless, the specific role of cyclins-CDK complexes in the global regulation of mitophagy needs be further evaluated.

### 2.3. E2F Transcription Factors and Autophagy

The adenoviral early region 2 binding factors (E2F) family of transcription factors, which is comprised of at least eight members (E2F1-8), are the downstream effectors of CDKs (Figure 1a). Their binding to pocket proteins RB, p107 also known as RBL1 and p130 also known as RBL2) are regulated by cyclin/CDK complexes, which phosphorylate pocket proteins, thereby causing the release of E2Fs from pocket proteins and the induction of E2F-dependent transcriptional activation. E2F targets include genes involved in the regulation of a myriad of cellular processes, such as the mitotic checkpoint, DNA-damage checkpoints, DNA synthesis and repair, differentiation, development, apoptosis, and metabolism [72,73,74,75,76]. E2F1 is also a transcriptional regulator of autophagy. Polager et al. demonstrated that in an inducible E2F system in human osteosarcoma (U-2 OS) cells, activation of E2F1 leads to the upregulation of the expression of crucial autophagy-related genes, such as *ULK1/ATG1*, *ATG5*, *DRAM*, and *MAP1LC3B/LC3B*, and enhances basal autophagy. In contrast, decreasing endogenous E2F1 expression using short hairpin RNAs (shRNAs) inhibits DNA-damage-induced autophagy [77]. In addition, the promoter region of Beclin-1 (encoded by BECN1), a key activator of the autophagy pathway, has been reported to be regulated by E2F transcription complexes [78]. Furthermore, the E2F downstream target BCL2/adenovirus E1B 19-kDa protein-interacting protein 3 (BNIP3) has also been shown to induce autophagy [79]. Moreover, a computational prediction from multiomics profiling of transcriptomes, proteomes, and phosphoproteomes during silica nanoparticles (SiNP)-induced autophagy in normal rat kidney cells identified *ATG9A*, *MAP1LC3B/LC3B*, UV radiation resistance associated gene (*UVRAG)*, and GABA type A receptor-associated protein (*GABARAP*) as transcriptionally regulated by E2F1. Although the exact mechanisms of how SiNPs activate autophagy are not clear, this study suggests that signaling through the CDK7-CDK4 axis potentiates SiNP-induced autophagy by phosphorylating pRB, activating E2F1 or FOXO3, and enhancing mRNA expression levels of several autophagy regulators and *ATG* genes [80].

Autophagy can be induced by the tumor suppressor transforming growth factor-β (TGFβ) in various cancer cell lines [81]. To be precise, TGFβ may regulate autophagy through pRB/E2F1-dependent transcriptional activation of multiple autophagy-related genes that function at various stages in the autophagy process. Indeed, the loss of E2F1 expression by siRNA significantly attenuates the TGFβ-mediated regulation of autophagy measured by impaired induction of Beclin-1, degradation of p62, and accumulation of LC3II. Together, these data suggest that E2F1 plays an important role downstream of TGFβ-mediated transcriptional activation of autophagy-related genes [82].

In addition to cancer, autophagy is associated with obesity; autophagy-related gene expression is upregulated in visceral fat in human obesity [83]. Interestingly, E2F1 expression is upregulated in the human adipose tissue of obese patients, which correlates with the increase in the expression of several key autophagic genes such as *ATG5* and *MAP1LC3B/LC3B*. Moreover, a direct binding of E2F1 to the MAP1LC3B promoter in adipose tissue explants has been demonstrated by chromatin-immunoprecipitation experiments. In cellular models, E2F1 overexpression in HEK293 cells triggered the promoter activity of autophagy-related genes and the autophagy flux and sensitized cells to tumor necrosis factor (TNF)-induced hyperactivation of *ATG12*, *MAP1LC3B*, and *DRAM1* promoter activity. Conversely, in mouse embryonic fibroblast-derived adipocytes from E2f1 knockout mice (E2f1−/−), autophagic gene expression and autophagic flux are downregulated in both basal and stimulated conditions [84].

However, the literature connecting pRB-E2F and autophagy is complex, as suppression of autophagy by E2Fs has also been reported. A study by Jiang et al. demonstrated that pRB restoration in a panel of pRB-deficient cells positively regulates autophagosome formation, maturation, and autophagy flux. In addition, they showed that increased pRB activity or lack of E2F1 trigger autophagy, and pRB-E2F interaction, is required for pRB-mediated autophagy. Moreover, excessive E2F1 activity blocked pRB-induced autophagy. One possible mechanism for how E2F1 mediates suppression of autophagy is the activation of Bcl-2 [44]. Bcl-2 has been previously revealed as a transcriptional target of E2F1 [85], which antagonizes autophagy by binding Beclin-1 and prevents activation of the PI3Kc3 complex [86]. This mechanism reveals a function of the pRB-E2F1 pathway that might contribute to its role in cancer suppression and resistance to cancer therapy.

Another suppressive function of the pRB-E2F pathway in the regulation of autophagy has been reported in ovarian cancer. Autophagy is one mechanism by which dormant, drug-resistant ovarian cancer cells survive in nutrient-poor environments. A frequently downregulated tumor suppressor gene in ovarian cancer, DIRAS3, has been illustrated to facilitate autophagy induced by amino acid deprivation in the presence or absence of serum. In fact, nutrient-depleted conditions lead to transcriptional upregulation of DIRAS3 by decreased binding of E2F1 and E2F4 and increased binding of CEBPα to the DIRAS3 promoter [87]. This mechanism was further confirmed by genetic and pharmacological inhibition of E2F1 and E2F4, which induced DIRAS3-mediated autophagy in ovarian cancer cells [88]. 

It is now evident that E2F1 transactivates more than 2000 genes and that the transcriptional regulation is highly contextual [89]. Therefore, further study will be required to determine the delicate balance between pRB and E2Fs in the regulation of autophagy. Nevertheless, in addition to E2Fs, pRB interacts with more than 200 proteins [90], including the hypoxia-inducible factor 1, that initiate the transcription of multiple genes involved in autophagy under hypoxic conditions [91]. Therefore, it will be necessary to distinguish between normoxic and hypoxic conditions when studying the regulation of autophagy by the RB-E2F pathway.

Collectively, previous works explicitly documented a prominent role of the pRB-E2F1 pathway in the regulation of autophagy (Figure 2c). Notably, in light of the numerous studies on this role of the pRB-E2F1 pathway, most of the previous studies on the roles of CKI and CDKs in autophagy (see the sections on cyclin-dependent kinase inhibitors and autophagy/cyclin-dependent kinases and autophagy) could be at least partially explained by modulation of the pRB-E2F axis and its pro-autophagic transcriptional program. Future studies must also address in more detail the relationships between pRB-E2F-induced autophagy and apoptosis. The elucidation of the molecular mechanisms by which the pRB-E2F pathway regulates cell fate through autophagy may provide a better understanding of how autophagy affects physiological and pathological processes.

## 3. Autophagy Modulates Cell-Cycle Regulators

While cell-cycle regulators interact with autophagic key players and modify autophagic processes, there is increasing evidence of a feedback loop of autophagy components that could influence the activity of cell-cycle regulators. Previously, mTOR inhibition through rapamycin has demonstrated the role of this complex in the regulation of the cell cycle, notably via the degradation of cyclin D1 and upregulation of p27^KIP1^ [92,93]. Additional later clues reinforce the importance of mTOR in cell-cycle regulation [94,95,96], indicating that autophagy is a potential determinant process modulating cell-cycle regulators. By regulating their availability, the autophagic program does not promote cell survival but enhances cell-cycle progression and may contribute to tumorigenesis.

### 3.1. Autophagy and Cyclin-Dependent Kinase Inhibitors

The protein p21^CIP1^ can be targeted by autophagy [97,98,99]. Under various conditions, inhibition of autophagy is able to stabilize p21^CIP1^ levels [97,98,99]. Of note, a recent study demonstrates that the ectopic expression of one of the master transcription factors of lysosomal biogenesis, TFEB, also induces p21^CIP1^ expression [100]. In addition, in the immune system, the expansion of immune CD8+ T cells requires autophagy to degrade specifically p27^KIP1^ [101]. Moreover, inhibition of multiple key elements of autophagy also increases the expression of p16^INK4A^ [45,46]. Indeed, while hepatic inhibition of autophagy through deletion of ATG5 increases the expression of p16^INK4A^ in liver [46], the loss of ATG7 enhances p16 level in satellite muscle stem cells [45]. Interestingly, an elegant study using time-lapse fluorescence microscopy and tracking an endogenous p16-mCherry reporter demonstrated that p16 accumulates in acidic cytoplasmic vesicles upon pro-autophagic stresses, including amino acid deprivation or oxidative stress [47]. 

Taken together, these data highlight the importance of autophagy in regulating p21^CIP1^, p27^KIP1^, and p16^INK4A^ levels, further impacting cell-cycle progression. While most of these studies were conducted in a non-tumoral context, these data are also in line with a pro-tumoral role of autophagy via the degradation of tumor suppressor proteins, such as CKIs (Figure 3).

### 3.2. Autophagy, Cyclin-Dependent Kinases and Cyclins

Both CDKs and cyclins are also targets of autophagy. For example, CDK1 is ubiquitinated by SCF^BTrCP^, enhancing its lysosomal degradation upon treatment with a DNA-damage chemotherapeutic agent, such as doxorubicin [102]. The CDK1 degradation is facilitated by the p62/HDAC6-mediated selective autophagy in breast cancer cells [103]. Cyclin D1, which is associated with CDK4 or CDK6, is frequently degraded by the modulation of autophagy. A first clue came out directly from the colocalization of cyclin D1 with Beclin-1 in glioblastoma recurrence [104]. Two more recent studies demonstrated that cyclin D1 is specifically degraded during autophagy in hepatocellular carcinoma and breast cancer [105,106]. In addition to cyclin D1, high-resolution live-cell imaging of breast cancer cells displayed ubiquitination of cyclin A2 in foci, further targeted by autophagy/lysosomal degradation [107]. 

Altogether, these results indicate that cell cycle regulators, including CKIs and CDKs-cyclin complexes, may be targeted by the autophagic process during cell cycle progression (Figure 3).

### 3.3. Mitophagy and Cell-Cycle Regulators

Notably, and similar to the role of autophagy, some studies display a role for mitophagy players in the regulation of cell-cycle regulators. Beyond the initiation step of mitochondrial fission, mitophagy is mastered by two key proteins, namely parkin and PINK1. Alterations in these proteins lead to impaired mitophagy and have been linked to cell-cycle arrest. Indeed, reduction of parkin results in cell-cycle arrest, and to some extent cellular senescence in primary lung, cochlear, or myocardial cells, via upregulation of p21^CIP1^ and p16^INK4A^ [108,109,110]. Mechanistically, the accumulation of dysfunctional mitochondria and abnormal ROS generation is suspected to participate in this phenotype [111]. Finally, genetic studies linked parkin-mediated effects with downstream CDKs: CDK6 [112,113], CDK1, and CDK2 [114]. Taken together, these studies underscore that the main mitophagic effectors modulate cell-cycle regulators. Nevertheless, the determination of whether this modulation consists of a direct targeting of cell-cycle regulators or is, rather, the result of multiple upstream molecular events, requires additional mechanistical studies.

## 4. Therapeutic Interventions Combining Cell-Cycle and Autophagy Modulators

The above-mentioned results suggest a strong reciprocal regulation between autophagy and cell-cycle regulators; perturbation of one leads to modulation of the other, as evidenced, for instance, by CDK4/6 or mTOR inhibitors. The importance of mTOR in cell-cycle regulation [94,95,96] reinforces the interest in using mTOR inhibitors in clinics as powerful anti-proliferative compounds, and for many cancer clinical trials as front-line therapy or alternative treatment to overcome resistance [115]. Unfortunately, faced with clinical data, either CDK4/6 or mTOR inhibitors are found to have clinical limitations as single agents in cancer therapy, as some patients develop clinical resistance. In addition, there is a lack of reliable biomarkers that can be used as prognostic indicators. Therefore, a combination therapies that provide additional therapeutic benefits is required. Interestingly, the interconnection between autophagy and cell-cycle regulators makes autophagy a promising co-target [56].

Chloroquine (CQ) and its derivative hydroxychloroquine (HCQ) are FDA-approved modulators of autophagy that inhibit lysosomal acidification and thereby impair autophagosome maturation. The effects of HCQ are explored in phase I and II clinical trials for various cancers. However, its use as a monotherapy does not provide satisfactory results [116,117].

Several preclinical studies have consistently shown that CDK4/6 inhibitors lead to autophagy activation in different cancer cells [55,118,119]^.^ For instance, palbociclib treatment leads to autophagy activation in breast cancer cells, which prevents palbociclib-induced senescence. However, combined usage of the CDK4/6 inhibitor palbociclib and the autophagy inhibitor HCQ results in growth arrest, accumulation of reactive oxygen species, and cellular senescence in cancers with an intact G1/S transition. In addition, this study identified that the Rb-positive and low-molecular-weight isoforms of cyclin E (cytoplasmic)-negative are reliable prognostic biomarkers in ER-positive breast cancer patients, and predictive of preclinical sensitivity to the combination of these drugs [56]. The synergistic effect of CDK4/6 and autophagy inhibitors has also been found in some solid cancers possessing a functional G1/S checkpoint, thereby suggesting a promising biomarker-driven antitumor strategy to treat breast tumors and other solid tumors [56,120]. 

In pancreatic ductal adenocarcinoma (PDAC), the co-encapsulate CDK4/6 inhibitor palbociclib and the autophagy inhibitor HCQ also generate synergistic antitumor effects in subcutaneous and orthotopic PDAC models [121]. Furthermore, a study of the malignant brain tumor glioblastoma multiform (GBM) combined the autophagy inhibitor, MPT0L145, with the CDK4/6 inhibitor abemaciclib. The results demonstrate significantly decreased cell proliferation, suppressed RB phosphorylation, and elevated ROS production, indicating the inhibition of autophagy by MPT0L145-sensitized GBM cancer cells to abemaciclib [122]. Furthermore, the efficacy of combination therapy is suggested by studies in acute myeloid leukemia (AML), i.e., the inhibition of CDK4/6 by abemaciclib or palbociclib and autophagy by CQ-suppressed cell growth and induced apoptosis in t(8;12) AML cells [123] and in a mouse xenograft model of t(8;12) AML [124] (Table 1).

These promising preclinical results led to several clinical trials (NCT04841148, NCT04523857, NCT04316169, and NCT03774472 [125]) to assess the safety and efficacy of cell-cycle regulators and autophagy inhibitors in breast cancer, as summarized in Table 2.

## 5. Conclusions

In constant interactions, cell-cycle regulators and modulators of autophagy are intrinsically related, facilitating the synchronization of the cell cycle and catabolism during cell-cycle progression. These observations explain most of the previous observations that link cellular senescence, stable cell-cycle arrest, and autophagy pathways. In addition to CKIs, CDKs, and cyclins, other cell-cycle regulators were found to participate in autophagy, as Aurora Kinase A [126,127] or p53 (comprehensively reviewed in [41]). Altogether, these studies reinforce the intimate link between the cell cycle and autophagy, adding another layer of complexity in the inter-regulation of these two processes.

The cell cycle and autophagy are two critical processes during tumorigenesis. In the tumorigenesis steps, cell-cycle and autophagy fluxes are coordinated (and especially conversely-correlated) during tumor progression and tumor therapeutic interventions (Figure 4). During these two processes, cancer cells stop proliferating and activate an autophagic program, leading ultimately to cell survival. In advanced stages of tumorigenesis, blocking the cell cycle appears to be required to sustain an important autophagic flux for cancer cells. This time-window to inhibit autophagy and overcome cancer cell survival will need further consideration.

Beyond the temporal modulation of autophagy during tumorigenesis (Figure 4), the importance of specifically and spatially targeting a tumor, rather than its microenvironment, remains crucial. Indeed, recent studies emphasized deleterious and pro-tumoral effects of modulating autophagy in tumor microenvironment, notably in immune cells and stromal cells [17,128]. For instance, autophagy inhibition may block cancer immunosurveillance, as its determinant to ensure tumor recognition and elimination [17]. Interestingly, not only autophagy modulators, but also cell-cycle blockers (including CDK4/6 inhibitors) are able to induce cellular senescence in normal and cancer cells [35,129,130]. Senescent cells display a specific senescence-associated secretory phenotype, which includes pro-inflammatory cytokines (IL-6, IL-8), growth factors (EGF, VEGF), and matrix metalloproteases (MMP-1, MMP-3) [131]. SASP contributes to local inflammation, establishing a pro-tumoral microenvironment contributing notably to cancer growth and tumor relapse [131,132,133,134]. Modifying autophagy and the cell cycle via induction of cellular senescence in the tumor microenvironment is thus an important factor to consider in future therapeutic-associated interventions. The use of senolytics (specific compounds eliminating senescent cells) or senomorphics/senostatics (specific agents limiting SASP), appears to be an important avenue to consider [135].

Overall, combining autophagy and cell-cycle modulators opens new therapeutic perspectives to initiate additional cell death mechanisms and synergetic antitumoral effects. Altogether, these studies reinforce the importance of timing, duration, and selective efficiency of drug administration, which are three main parameters to optimize for autophagy-/cell-cycle-related therapeutic interventions in the future.

## Figures and Tables

**Figure 1 cancers-14-00153-f001:**
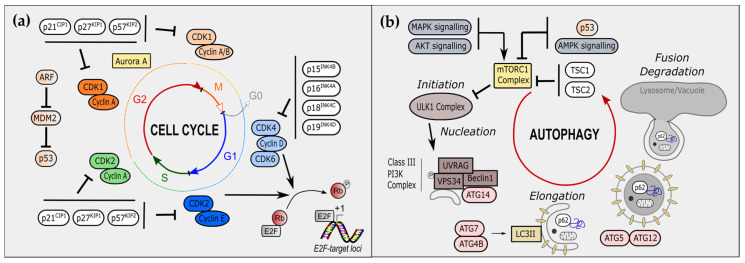
Cell-cycle regulators and autophagy players. (**a**) Overview of the main cell-cycle regulators. The cell cycle is divided into four steps: G0 (grey)/G1 (blue), S (green), G2 (red), and M (orange) phases. Cyclin-dependent kinases (CDKs) associate with cyclins along the cell cycle, allowing its progression. In early G1, CDK4 and CDK6 bind to cyclin D. In late G1, CDK2 is associated with Cyclin E. CDK4/6-cyclin D and CDK2-cyclin E phosphorylate RB leading to E2F release and transcription at E2F-target loci of cell-cycle associated genes. Cyclin A associates with S-phase CDK2 and G2/M-phase CDK1. CDKs are inhibited upstream by cyclin-dependent kinase inhibitors (CKIs). In G1 phase, INK family including p15^INK4B^, p16^INK4A^, p18^INK4C^ and p19^INK4D^, inhibits CDK4/6-cyclin D. In other phases, CIP/KIP family including p21^CIP1^, p27^KIP1^ and p57^KIP2^. Additional cell-cycle regulators include the Aurora Kinase A and the ARF/MDM2/p53 axis. (**b**) Overview of steps and actors of autophagy, called macro-autophagy. Initiation is mediated by the ULK1 complex, which mediates phosphorylation of VPS34 and formation of phagophore through nucleation and the association of UVRAG, Beclin-1 and ATG14. Elongation of phagophore is allowed by LC3II formation, by ATG7 and ATG4B, and LC3II association, by ATG5 and ATG12. Proteins can be also specifically targeted to phagophore by p62. Organelles, such as mitochondria, can be Fusion of phagophore with lysosome constitutes the last step, ultimately followed by degradation of internal components.

**Figure 2 cancers-14-00153-f002:**
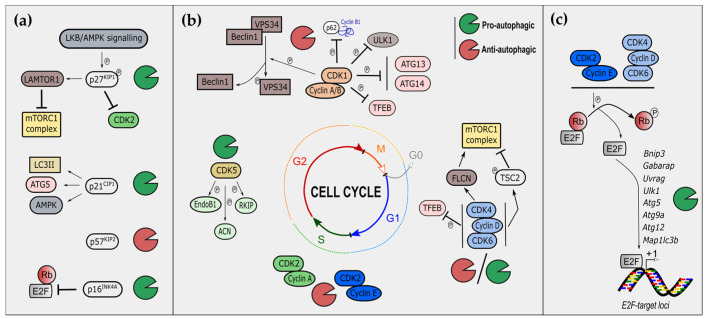
Regulation of autophagy by cell-cycle regulators. (**a**) Regulation of autophagy by cyclin-dependent kinase inhibitors (CKIs). Phosphorylation of p27^KIP1^ by the LKB/AMPK signaling pathway participates in the recruitment of LAMTOR1 to lysosome and the subsequent inhibition of mTORC1, resulting in autophagy induction; p27^KIP1^ is also pro-autophagic through CDK2 inhibition, while p21^CIP1^ is pro-autophagic through mechanisms involving LC3II interaction, ATG5 stabilization, and AMPK phosphorylation. P57^KIP2^ is anti-autophagic with unknown clear mechanisms. p16^INK4A^ mediates autophagy through the RB-E2F axis. (**b**) Regulation of autophagy by cyclin-dependent kinase (CDKs). CDK1 phosphorylates multiple key autophagic players, including VPS34, p62, ULK1, ATG13, ATG14, and TFEB, in inhibiting global autophagy. CDK4 phosphorylates folliculin (FLCN) modulates mTOR recruitment at lysosomal surface. CDK4-6 can also phosphorylate TSC2 to inhibit the mTORC1 complex. CDK5 phopho-substrates include Endophilin B1 (EndoB1), Acinus (ACN), and RKIP promoting autophagy. (**c**) Regulation of autophagy by RB-E2F axis. E2F target loci include many autophagy-encoding genes, namely *BNIP3, GABARAP, UVRAG, ULK1/ATG1, ATG5, ATG9A, ATG12,* and *MAP1LC3B*.

**Figure 3 cancers-14-00153-f003:**
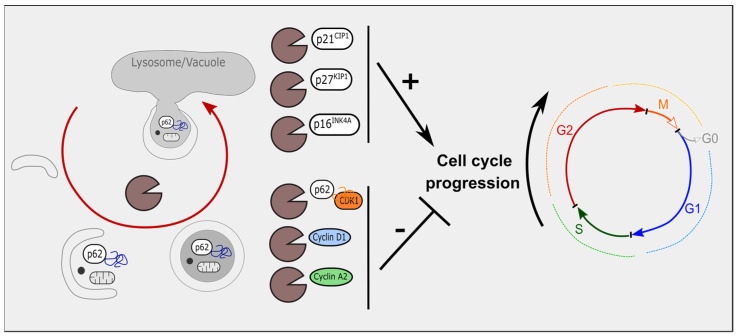
Cell-cycle regulators are targets of autophagy. p21^CIP1^, p27^KIP1^ and p16^INK4A^ are among the three CKIs targeted by autophagy, their degradation ultimately leading to the progression of the cell cycle. Cyclins and CDKs are also degraded by autophagy. CDK1 binds to p62. Cyclin D1 and cyclin A2 are degraded by macroautophagy. Altogether, degradation of CDKs and cyclins block cell-cycle progression.

**Figure 4 cancers-14-00153-f004:**
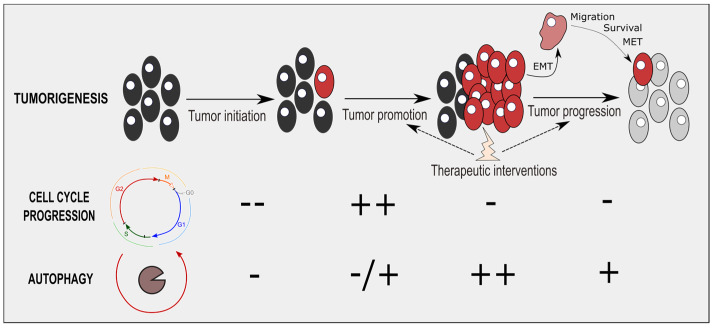
Cell-cycle and autophagic processes during tumorigenesis. During tumor initiation, the cell cycle and autophagy are mostly downregulated. Autophagy is then antitumoral initially, limiting this tumor initiation step, notably via the degradation of oncogenic molecules. During tumor promotion, cell-cycle progression is reactivated while autophagy is partly enhanced. Tumor progression, notably through epithelial-mesenchymal transition (EMT), migration, survival, and mesenchymal-epithelial transition (MET), reduces drastically cell-cycle progression, while enhancing autophagic fluxes. Resister cells of therapeutic interventions display cell-cycle arrest and autophagy induction.

**Table 1 cancers-14-00153-t001:** Combination therapies of cell-cycle inhibitors and autophagy inhibitors. CQ: chloroquine; HCQ: hydroxychloroquine.

Intervention	Cancer Type	Refs
– CDK4/6 inhibitor: palbociclib – Autophagy inhibitor: HCQ	Breast cancer, solid tumors	[56]
– CDK4 inhibitor: 2-bromo-12 and 13-dihydro-5 H-indolo-dione [2 and 3-a] pyrrolo [3 and 4-c] carbazole-5 and 7 (6H)-dione – Autophagy inhibitor: CQ	Solid tumors	[120]
– CDK4/6 inhibitor: palbociclib – Autophagy inhibitor: HCQ	Pancreatic cancer	[121]
– CDK4/6 inhibitor: abemaciclib – Autophagy inhibitor: MPT0L145	Brain cancer (glioblastoma multiforme)	[122]
– CDK4/6 inhibitors: abemaciclib and palbociclib – Autophagy inhibitor: CQ	t(8;21) Acute myeloid leukemia	[123]
		[124]
– CDK4/6 inhibitor: palbociclib – Autophagy inhibitor: HCQ	Estrogen receptor-positive, HER2-negative metastatic breast cancer	[125]

**Table 2 cancers-14-00153-t002:** Clinical trials in breast cancer patients with CDK4/6 and autophagy inhibitors. Information obtained from ClinicalTrials.gov in December 2021. HCQ:hydroxychloroquine.

Clinical Trial	Official Title	Condition	Intervention	Study Description	ClinicalTrials.gov Identifiers
PALAVY	A phase II trial of Avelumab or hydroxychloroquine with or without palbociclib to eliminate dormant breast cancer	Early-stage ER + breast cancer	HCQ, Avelumab, palbociclib	Randomized, open label phase II clinical trial that will assess the safety and early efficacy of hydroxychloroquine or Avelumab, with or without palbociclib, in early-stage ER+ breast cancer patients who are found to harbor disseminated tumor cells (DTCs) in the bone marrow after definitive surgery and standard adjuvant therapy.	NCT04841148
ABBY	A phase II pilot trial of abemaciclib or abemaciclib and hydroxychloroquine to target minimal residual disease in breast cancer patients	Invasive breast cancer	abemaciclib HCQ	Randomized, open label phase II clinical trial that is testing whether the use of hydroxychloroquine and abemaciclib can reduce the number or eliminate DTCs in bone marrow.	NCT04523857
	Hydroxychloroquine, abemaciclib, and endocrine therapy in hormone receptor positive (HR+)/Her 2 negative breast cancer	HR+/Her 2- advanced breast cancer, advanced solid tumors	Abemaciclib HCQ	Non-randomized, open label phase I clinical trial that assessed safety, tolerability, and efficacy of abemaciclib combined with the autophagy inhibitor hydroxychloroquine in advanced solid tumors and HR+/Her 2-advanced breast cancer	NCT04316169
	Phase I/II safety and efficacy study of autophagy inhibition with hydroxychloroquine to augment the antiproliferative and biological effects of pre-operative palbociclib plus letrozole for estrogen receptor-positive and HER2-negative breast cancer	HR+/HER2-bbreast cancer	HCQ, letrozole, palbociclib	Open label phase I/II clinical trial investigating the side effects and best dose of hydroxychloroquine when given together with palbociclib and letrozole before surgery in treating patients with estrogen receptor positive, HER2-negative breast cancer.	NCT03774472

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
