# Peer review of "The Intricate Interplay between Cell Cycle Regulators and Autophagy in Cancer"

_cancers, 2021, doi:10.3390/cancers14010153_

Round 1
Reviewer 1 Report
The manuscript entitled "The intricate interplay between cell cycle regulators and autophagy in cancer." has been submitted as a review ba Ziegler et al.
The review deals with the interplay of cell cycle regulation and autophagy in the context of cancer. The review is comprehensive and informative.
I have only suggestions concerning the figures. Some of the details, like in Figure 1 or Figure 3, are very small and faint and therefore difficult to see. It would be helpful to improve this aspect.
Author Response
Thank you for your valuable suggestion. Please see the attachment.

Reviewer 2 Report
In this manuscript Dorian V. Ziegler and colleagues highlight the relationships between cell cycle regulators, including Cyclin-dependent kinase inhibitors, Cyclin-dependent kinase and E2F factors in the control of autophagy all along cell cycle in cancer context.
Authors give a detailed description of the different steps of cell cycle and how regulators of cell cycle modulate autophagy. Furthermore, they also report how autophagy itself modulates cell cycle.
The figures are well done and very informative (I only recommend to provide images at high resolution). References are update to the most recent discoveries.
Following, I will indicate some suggestions that (in my opinion) may help not only to improve the manuscript, but also to capture a greater audience of readers.
- Authors should insert a section where they describe the autophagy mechanisms and molecular mechanism. Hence, the review could be also read by persons that are not expert in the field of autophagy. I also recommend adding the figure 1 b to this new paragraph.
- Authors in the abstract report: “this review opens therapeutic perspectives in modulating together these two aspects to block cancer progression.” I noticed that some therapeutic opportunities are reported along the manuscript. However, to facilitate the reader, they should be included in a separated section with a dedicated table, in which the reader can easily found the relative references and tumor type involved.
- Several publications account for mitochondrial homeostasis (and mitophagy, especially) an important role in cell cycle. Since the review focus on autophagy mechanisms, authors should also report the diverse selective autophagy mechanisms involved in regulation of cell cycle.
- Is there a specific relationship between autophagy and cell cycle in the different steps of tumorigenesis (such as tumor development, growth, and metastasis)? If yes, authors should provide evidences regarding this.
- Again, is there a relationship between autophagy-cell cycle and inflammation/tumor microenvironment? If yes, authors should provide evidences regarding this.
- Finally, an abbreviation list could be useful.
Minor points:
Please correct the first part of the title. I think there is a typo.
Please insert the number/symbol corresponding to the affiliation in the authorship (if necessary)
Author Response

(The authors gave the same response as above.)

Reviewer 3 Report
In this review the authors aim to dissect the molecular mechanisms underlying the interplay between cell cycle and autophagy in cancer. It is well known that both cell cycle and autophagy dysregulation are key events that take part to oncogenic transformation as well as in later steps of tumor progression, and the idea to study and detail how these processes are interconnected represent a very fascinating field that deserves to be deepened.
The manuscript is very well written and of interest for the community. However, I think that the paper could be improved with some additional references and discussion about novel players involved in cell cycle and autophagy interplay and I recommend address the following points that could improve the manuscript.
- Correct the error type in the first word of the title
- Line 39: correct “fectors”
- To extend the discussion about the interconnexion between cell cycle and metabolism, the introduction would benefit of a mention about PDK, an important enzyme involved in cell cycle regulation (doi: 10.1016/S0960-9822(00)00801-0; 10.1074/jbc.M802589200) that has been reported to contribute to glycolytic regulation. The link is reported in this important paper that the author should cite doi 10.1016/j.stem.2012.10.011
- Although some regulators could act as autophagy inducers and other ones as inhibitors, it is clear that cell cycle sustains the anabolic pathway towards a cell growth/cell proliferation program. The authors should put more emphasis on cell cycle regulation by mTOR since is strictly linked with autophagy doi: 10.1007/978-1-4939-0888-2_7; doi: 10.1128/MCB.24.1.200-216.2004. In this sense, it is important to discuss that mTOR inhibitors, that are often used in clinic, also lead cell cycle arrest doi: 10.1158/1078-0432.CCR-13-3172
- Since autophagy modulators have been exploited in anticancer research (doi: 10.3389/fphar.2021.650559) it would be interesting to investigate if there are some drug employed in cell cycle inhibition that may also serve to modulate autophagy and hence, their combination with autophagy inductors or inhibitors could enhance their effect. This point should be discussed.
- In the figure 2, I recommend change the colors for pro-autophagy and anti-autophagy symbols; I suggest put in green “pro-autophagy” symbol, and in red “anti-autophagy”. This would improve the visive readability.
- For more completeness I recommend include CDK4/6 as regulator of TFEB and lysosomal biogenesis doi: 10.1083/jcb.201911036. Also, TFEB has been described as a modulator of p21/WAF1/CIP1 during the DNA Damage Response, I recommend include this reference doi:10.3390/cells9051186
Author Response

(The authors gave the same response as above.)

Round 2
Reviewer 2 Report
Dear authors,
I would like to thank you for having answered to all my concerns.
The manuscript has been significantly improved and I consider it ready for publication.
Reviewer 3 Report
The authors addressed to my concerns, the manuscript has been improved and can be accepted.